# Effects of Transport and Lairage on the Skin Damage of Pig Carcasses

**DOI:** 10.3390/ani10040575

**Published:** 2020-03-29

**Authors:** Bert Driessen, Sanne Van Beirendonck, Johan Buyse

**Affiliations:** 1Research Group Animal Welfare, 3583 Paal, Belgium; bert.driessen@dierenwelzijn.eu; 2Laboratory of Livestock Physiology, Department of Biosystems, KU Leuven, 3001 Heverlee, Belgium; 3Bioengineering Technology TC, KU Leuven, 2440 Geel, Belgium; sanne.vanbeirendonck@kuleuven.be

**Keywords:** animal welfare, pigs, skin damage, transport

## Abstract

**Simple Summary:**

Transport and lairage conditions can negatively affect pig welfare and carcass quality. The effects of transport and lairage conditions are not clear because some studies were focused on one or two main factors within a well-controlled environment, without considering the interaction of commercial conditions. Therefore, the aim of the study was to assess pig welfare at the slaughterhouse based on the presence of skin damage and to evaluate the effects of transport and lairage conditions. For research purposes, 4507 pigs were transported from one farm to one slaughterhouse. On the slaughter line, skin damage was separately scored per carcass part (shoulder, middle, and ham). The incidence of skin damage was most prevalent in the shoulder region of the pig carcass. Sex, wind velocity, regrouping, transport combination, transport compartment, lairage time, and ham angle affected the skin damage incidence. In conclusion, the incidence of skin damage is influenced by many factors and is an indicator of the level of welfare exercised during transport and slaughterhouse conditions.

**Abstract:**

Transport and associated handling can have adverse effects on pig welfare. The transport of fattening pigs can cause economic losses by virtue of mortality, skin damage, and the general deterioration of meat quality. A total of 4507 fattening pigs were transported from a farm to a commercial slaughterhouse (distance 110 km) in 128 transports. Skin damage was visually assessed in the slaughter line in different parts of the carcass, i.e., shoulder, middle, and ham, using a 4-point scale. The incidence of skin damage was most prevalent (31%) in the shoulder region of the pig carcass. Sex, wind velocity, regrouping, transport combination, transport compartment, lairage time, and ham angle affected the skin damage incidence. In conclusion, scoring the incidence of skin damage is an indicator of the level of welfare exercised during transport and the slaughterhouse conditions. Furthermore, skin damage monitoring can be used to determine critical control points in the transport procedure. Given the importance from both a commercial and welfare perspective, it should be a powerful incentive to handle fattening pigs with care during the transport process and the lairage period.

## 1. Introduction

Animal welfare issues are of growing societal and scientific concern. From the animal’s point of view, transport is a very complex and stressful event [1,2]. Transport and associated handling during transport and lairage can have adverse effects on pig welfare [3,4,5]. These adverse effects are directly related to psychological, physical, environmental, and metabolic factors. When physiological control systems, which maintain homeostasis, are overtaxed, the term stress is used [6]. The transport process of fattening pigs, i.e., from loading till slaughter, can cause economic losses by virtue of mortality, skin damage, and the general deterioration of meat quality [7,8]. However, the extent of such economic losses will ultimately depend on the intensity and duration of the stress implied and the susceptibility of the animals to stress. Injuries are accepted as a major and non-controversial indicator of poor welfare [9]. Therefore, skin damage can act as an animal welfare indicator and reflect the quality of the animal’s social and physical environment [10,11]. Two main causes of skin damage can be stated. Firstly, an unadapted social environment can cause excessive fighting (e.g., after mixing), ultimately resulting in the occurrence of skin damage. Secondly, a poorly designed physical environment can also lead to skin damage by virtue of physical trauma [12]. Moreover, mounting due to, e.g., overstocking can induce skin damage. According to Dalmau et al. [13], the source of the skin damage (fighting, rough handling, and mounting) can be correlated with the anatomical location on the body. According to the source of the skin damages, differences in incidences on these parts can be expected. Therefore, it is interesting to score the skin damage per body part [14].

According to EU regulation [15], large slaughterhouses slaughtering more than 1000 animal units per year should be able to document animal welfare. Compliance with these requirements necessitates the development of on-site tools for the continuous monitoring of the welfare of slaughter pigs, e.g., the monitoring of skin damage. Besides animal welfare implications, skin damage also causes serious commercial problems by decreasing the grade and, in turn, the value of the carcass [16]. In addition, the presence of a haematoma in the underlying tissue and its negative influence on meat quality must also be taken into account [16]. For example, in Italy, high-quality hams (“Parma hams”) must be completely unblemished [14]. 

Transportation often coincides with a change in ownership whereby responsibility for the animal’s welfare may be compromised. However, the welfare of animals during their transport is the joint responsibility of all people involved [17]. Holding people accountable for losses is a great motivator to prevent losses. The feedback of the results of skin damage from the slaughterhouse to both the pig farmer and the transporter is useful to find and reduce the cause of skin damage in pig transports [18]. Skin damages can be attributed to the infrastructure of the pig farm, the way of driving, as well as to the infrastructure and condition of the truck [19]. If the problems are not tackled, price reductions can be imposed or purchase can no longer be guaranteed. In addition, penalties may be imposed by animal welfare inspectors of the government or slaughterhouse organisations [20]. Furthermore, slaughterhouse organisations can give penalties or rewards [18].

Past research has demonstrated the diversity of influencing factors concerning skin damage, making the appearance of skin damage a multifactorial problem [4]. Most of the previous studies only focused on one or two important factors within a well-controlled environment, without considering the interaction of commercial conditions. In studies, influences of sex, weight, regrouping, poor handling, transport conditions, and lairage conditions in relation to skin damage have been investigated [4,5,16,21]. Hence, the relative importance of influential conditions is not always apparent. Therefore, the objective of this study is to consider and highlight the possible effects of different factors, from transport to slaughter, on the incidence of skin damage on pig carcasses.

## 2. Materials and Methods 

### 2.1. Animals and Housing

A total of 4507 hybrid pigs (Piétrain × Hypor), being heterozygous for the halothane gene, were used. Both female pigs and castrated males were raised in the same housing conditions at the Zootechnical Centre—KU Leuven R&D (ZTC). All pigs were individually marked with an ear tag number before weaning. At the moment of the marking of the piglets, the sex was registered. The fattening period started at about 22 kg, and pigs were kept within the same group composition on concrete slatted floors during fattening. The minimal floor space provided for each pig was 1 m^2^, as opposed to the minimum prescribed required space of 0.65 m^2^ for a pig of 85–110 kg [22]. All pigs had ad libitum access to water and a commercial diet. Management and control was based on the “all in—all out” principle for each room.

The day before slaughter, the pigs were weighed individually with an electronic weighing scale and marked with an individual tattoo number on each side of the body. This number allows individual identification of the carcasses in the slaughterhouse. Pigs were fasted 16 h before transport.

During the study, continuity was maintained with regard to the technicians who took care of the animals, to exclude the effect of handling during the fattening period on the occurrence of skin damage on the carcasses [16]. Slaughter dates were registered, so that possible seasonal effects could be taken into account. Animals were treated in accordance with the regulations of the Council Directive 86/609/EEC regarding the protection of animals used for experimental and other scientific purposes [23].

### 2.2. Transport and Slaughterhouse

The following transport procedures were standardized. On the day of slaughter, all pigs were loaded on the trailer by the truck driver and handlers of the ZTC using light-weight driving boards and a tail gate lift. Pigs were not treated with sedatives in accordance with the Belgian Royal Decree of 8th September 1997. The pigs were transported in 8 pens in two-tiered trailers (each tier consists of 4 compartments), it was strived for 12 pigs per pen (Figure 1). The pigs were mainly transported in the upper tier of the trailers. The pigs were always transported before noon. The distance from the Zootechnical Centre (Lovenjoel, Belgium) to a commercial slaughterhouse (Comeco, Belgium) was 110 km. The floor space of the pens on the trailers was 6.63 m^2^ (trailer 1) or 4.98 m^2^ (trailer 2) (Table 1). The loading density did not exceed the maximum loading density (235 kg/m^2^) of the EC Regulation [20], so that pigs were able to sit or lie down during transport. It was decided to transport the pigs in both mixed (during loading) and unmixed groups, in the ratio of mixed: 4169 pigs in 111 transports and unmixed: 338 pigs in 17 transports. The unmixed treatment consisted of keeping pigs in the same social group during fattening, transport, and lairage. The transport combinations utilised, i.e., the number and specified combination of truck, trailer, and driver are mentioned in Table 1. The trucks and trailers chosen differed in age and construction firm. Vents for natural ventilation stretched the length of each trailer, on both sides and on each tier.

All pigs were transported to the same slaughterhouse using the same route. Each group of pigs was unloaded as soon as possible after the arrival at the slaughterhouse. In the lairage, all groups of pigs were kept in pens (2.00 × 3.07 m) in the same groups as during transport. In the lairage, all pigs were showered by a water sprinkling system. The lairage facility was supplied with drinking nipples. The time spent between unloading the pigs and the moment pigs were driven to the stunning area, has been defined as the lairage time, and is recorded as such. The pigs were stunned with a Midas-stunning device (Stork MPS, Lichtenvoorde, The Netherlands). Head-to-back electrical stunning (240 V, 800 Hz for 2 s), which induces cardiac arrest, was applied. 

### 2.3. Data Collection

#### 2.3.1. Weather Characteristics during Transport

Climate characteristics were measured during each transport. The dry air temperature and humidity sensors (Miravox, Stabroek, Belgium) were set in the middle at 3 cm under the ceiling in each compartment of a trailer. Data on the wind velocity were provided by the Royal Meteorological Institute of Belgium (Brussels). The temperature-humidity index (THI) is calculated by combining temperature and humidity using the method reported by Ravagnolo et al. [24]: THI = (1.8 × T + 32) − ((0.55 − 0.0055 × RH) × (1.8 × T − 26))(1)
where T = air temperature (°C) and RH = relative humidity (%).

The slaughter dates were registered so that possible seasonal effects could be taken into account. The seasons wherein the pigs were transported to the slaughterhouse were defined as groupings of three whole months, as identified by the Gregorian calendar: spring (21 March–20 June), summer (21 June–20 September), autumn (21 September–20 December), and winter (21 December–20 March).

#### 2.3.2. Time Sampling during Transport and Lairage

The transport time is defined as the time spent between departure from the farm and the arrival at the slaughterhouse and was registered each transport. The duration of unloading, defined as the time spent between arrival at the slaughterhouse and the unloading, was always monitored and recorded.

#### 2.3.3. Carcass Variables

Thirty-five minutes post-mortem, carcasses were graded with a SKGII-device (Schlachtkörper Klassifizierungs Gerät, Tecpro GmbH, Willich, Germany), which combines 4 physical measurements (ham angle, ham width, loin width and back fat thickness) to estimate the lean meat content. At that moment the carcass was weighed.

#### 2.3.4. Skin Damage

The skin of the left carcass side of each pig transported (*n* = 4507) was evaluated in the slaughter line 30 min post-mortem by 2 scorers separately (always the same 2 persons). After scalding and evisceration, skin damage were visually assessed in different parts of the carcass, i.e., shoulder, middle and ham, using the anatomical locations and procedures described by Barton Gade et al. [14]. A 4-point photographic scale was used, starting at 1 = no damage, 2 = slight skin damage, 3 = skin damage affecting quality, and 4 = extreme damage (Figure 2). The used scale can be considered as a combination of the number and the severity of the lesions. Only recent/fresh skin damage was registered. The change in colour from red (recent lesion) to yellow (older lesion) offers the possibility to differentiate between recent and older skin damage [25]. Fresh skin damage may indicate damage due to transport and lairage conditions, e.g., overcrowding and poor handling [18]. Skin damage was assessed per body part, but the composite score always reflects the highest value and is called the ultimate score [14]. The ultimate score can be used to divide the carcasses in classes of acceptable and unacceptable skin damage [26].

### 2.4. Statistical Analyses

Pigs were considered as the experimental units because the data on meat quality were collected for each individual pig. The data were analysed with a proportional odds model (GLIMMIX Procedure of SAS 9.4) [27], with skin lesion as the response variable and transport number as a random factor (pigs are grouped per transport number). Statistical significance was accepted at *p* < 0.05. Ham angle had the highest correlation with the skin damage and hence ham angle as carcass conformation parameter is included in the model. The first step in the statistical analyses involved screening of all single registered variables (sex, transport season, weather characteristics, regrouping or not during loading, transport combination (truck, trailer and driver), transport compartment, transport conditions (loading density, transport duration, and unloading time), the duration of lairage, and carcass conformation). Therefore, each of the independent variables was separately introduced as a fixed effect in the model. Variables with a significant value (*p* < 0.05) were selected for the multiple model. In the second step, a backwards elimination of variables was performed to analyse variables simultaneously. Therefore, a model was built per carcass location (shoulder, middle, and ham) with the transport number as a random factor. Again, only factors significant at the 5% level were kept in the model. 

## 3. Results

An overview of the weather, transport, and carcass variables is presented in Table 2. Large variations (minimum versus maximum, Table 2) in transport, unloading, and lairage time were determined. 

Thirty-one percent of the carcasses had shoulder damage (Table 3). This proportion is higher than the percentage of carcasses with middle (28.26%) and ham (11.78%) damage. Most damaged carcasses (score > 1) had slight skin damage, i.e., score 2 (Table 3). Extreme skin damage (score 4) was found on the shoulder region (2.32%), the middle (0.77%), and on the ham (0.35%) of the carcasses. 

The incidence of skin damage is influenced by seven variables, i.e., sex, wind velocity, regrouping, transport combination, transport compartment, lairage time, and ham angle (Table 4). Firstly, the sex of a pig was a determinant to damage in the shoulder region. Castrated males suffered less damage on the shoulder (*p* = 0.00035) (Table 5). In contrast, the middle of the carcasses of female pigs was slightly more damaged than in the shoulder region (*p* = 0.00038) (Table 5). Sex did not affect ham damage. Secondly, wind velocity did influence shoulder, middle, and ham damage. During transport with low wind velocity, carcasses displayed more damage on the shoulder region (*p* = 0.0009), middle (*p* < 0.0001), and ham (*p* = 0.0009). Thirdly, the analysis showed an effect of regrouping pigs just before transport. Mixing pigs caused more skin damage on shoulder (*p* < 0.0001) and middle (*p* = 0.0267) (Table 5). Fourthly, the transport combination played a significant role in the incidence of skin damage, both at the level of the shoulder, the middle, and the ham. The least skin damages were seen in transport combination 2. In contrast, combination-4-transported pigs had the the most skin damages. Fifthly, pigs transported in the lower tier show higher damage scores on the ham than pigs transported on the upper tier. Sixthly, longer lairage time resulted in pigs with more damage to the middle (*p* = 0.0017) and ham (*p* = 0.0483). Lairage time varied during this study. Although a lairage time of 2 hours after each transport was the aim, in practice, the mean lairage time was 83.8 ± 34.0 min (Table 2). Finally, ham angle effected carcass damage. The carcasses of well-conformed pigs displayed more damage on the shoulder region (*p* = 0.0004). Transport season, THI, transport time, and transport density did not influence skin damage on the shoulder, middle, or ham.

## 4. Discussion

### 4.1. Skin Damage Incidence

Geverink et al. [28] also found a higher incidence of skin damage on the shoulder region than on the middle or ham part. Agonistic encounters are indicated as one of the main causes. Normally, during fighting, bites are targeted mainly at the ears, face, neck, and shoulder [29]. The presence of any skin damage (scores 2, 3, or 4) on the ham has serious detrimental economic consequences, given that this part is destined mainly to be processed into dry-cured ham [25]. Our findings of severe damage (2.32% on shoulder part) are between the percentages in the UK (4.0%) and Spain (1.5%), as mentioned by Gispert et al. [30]. According to Faucitano [16], scores 3 and 4 are considered unacceptable. This would mean that in our study 85.14% (= scores 1 and 2) of the carcasses should be acceptable. However, even score 2 may be considered unacceptable for sensitive markets. For our study, it would mean that only 53.78% (= score 1) of the carcasses are acceptable for these sensitive markets.

### 4.2. Influencing Parameters

The incidence of carcass damage is influenced by seven variables, i.e., sex, wind velocity, regrouping, transport combination, transport compartment, lairage time, and ham angle (Table 4). 

Firstly, there was an effect of sex, females vs. castrated males. In accordance with Warriss and Brown [31] and Mota-Rojas et al. [32], castrated males showed a higher incidence of skin damage at the slaughter line. Warriss et al. [33,34] concluded that a cause of skin damage is the result of pre-slaughter fights amongst pigs, and these take place to a higher degree among males than among females. 

Secondly, the incidence of skin damage is influenced by wind velocity. Pigs are sensitive to high ambient temperatures [35,36], and since pigs cannot sweat, they must rely on other means of thermoregulation. In a high ambient temperature and high pig density, pigs become irritated because they cannot find a place to cool down. Finally, this leads to increased aggression [37]. A higher wind velocity can help the pigs to cool down.

Thirdly, the analysis showed an effect of mixing pigs just before transport. Unfamiliar animals will fight after mixing to establish dominance in a social hierarchy. Fighting does not only result in an increased incidence of skin damage, but also impede resting behaviour [38]. In general, fighting amongst pen mates does not occur during transport [39], but rather in lairage when they seek to assert social dominance. Damage generated by aggressive behaviour is related to two different behavioural patterns. The damage of the anterior third of the body results from attempts to target the head, neck, and shoulder [28]. Wounds caudal of the body are related to mounting. The back will be damaged with the claws of the forelimbs when pigs mount one another and scratch. Mounting damage usually occurs in overcrowded conditions in pens, at loading and unloading, or on route to the stunning chutes [16]. Several other studies also showed that mixing increased the frequency of skin damage [40,41] caused by aggression with a view to establish social dominance. However, in practice, pigs are regrouped before transport to achieve uniform groups of weight for slaughter and during lairage at the slaughterhouse, to fill the lairage pens to maximal capacity [30]. To reduce the negative impact of such regrouping on the welfare of the pigs and the occurrence of skin damage, trailer decks and lairage pens should be equipped with mobile dividers in order to keep pigs in smaller contained batches, without the need to regroup pigs of different groups [38]. The regrouping of pigs and the associated occurrence of fighting in a pen also has detrimental effects on pigs resting in neighbouring pens as they are disturbed by the vocalisation of fighting pigs [42].

Fourthly, the combination of transportation factors played a significant role in the incidence of carcass damage, for example, the driving behaviour and the construction of the transporter. In our study, the transport combination of 1 and 2 differs only in driver, but the same truck and trailer have been used, but depending on the body part there are 8% to 17% more carcasses without injuries. Driving style can be responsible for the difference in skin damage. Rough driving, which comprises high acceleration and hence violent braking, may cause postural instability [43], toppling, sliding, and excessive corrective muscular action, resulting in bruising, muscular fatigue, fear, and general injuries to the animals. According to EC Council Directive 1/2005 [20], farm animal transporters and handlers have to attend a training programme to obtain a certificate of professional competence, which came into force from the 5th of January 2008. A training programme for animal transporters and handlers should minimize animal transport stress and optimize animal welfare [44].

Fifthly, the pigs’ transported in the lower tier show higher damage scores on the ham than pigs transported on the upper tier. This is in line with the findings of Barton Gade et al. [14]. However, they described effects on the shoulder and middle part. Barton Gade et al. [14] suggest that the higher vibration and noise encountered in the lower tier are possible sources for this higher incidence. Moreover, the height of the lower tier was lower than the height of the upper tier. This can result in a less ventilation in the lower tier, which undoubtedly contributes to the results [14].

Sixthly, lairage time varied during this study. The supply of pigs was sometimes insufficient for a number of hours, causing the number of pigs in lairage to be depleted. As a consequence, new pigs arriving at the slaughterhouse were driven to the stunning area immediately, or after a short lairage time. A longer lairage time resulted in pigs with more damage on the shoulder, middle, and ham, which is in accordance with Warriss et al. [34]. Reported skin damage was seen to be increased when pigs were kept in lairage for 3 h, as compared to lairage for 1 h. Nanni Costa et al. [45] found that pigs staying in lairage for 2 h suffered less skin damage compared to pigs held in lairage for 22 h. Longer resting time often causes a considerable increase in the degree and extent of fighting, especially when unfamiliar groups are mixed [34]. Clearly, a longer lairage time allows the pigs to rest, but increases the risk of aggression and thereby the incidence of skin damage [46]. Geverink et al. [28] found a negative association between lairage time and skin damage in Dutch slaughterhouses. This was in contrast with Belgian slaughterhouses where they could not find such an association. The remarkable differences between the findings of the Dutch and Belgian part of the study of Geverink et al. [28] are probably caused by the utilisation of tranquillizers to transport animals during their study in Belgium, resulting in generally low levels of agonistic behaviour. Since September 1997, the use of tranquillizers to transport animals to the slaughterhouse is forbidden. Thus, although a lairage should allow the pigs to recover form transport stress, it can be a source of stress, which can result in more skin damage [46].

Finally, the individual conformation influences the incidence of skin damage on the ham. Several studies [47,48] have reported that the heaviest pigs within a group fight more and the largest pigs are more likely to win [49]. Olesen et al. [50] detected a positive correlation (r = 0.35) between live weight and aggressive behaviour. On the other hand, it could be that well-conformed pigs have more sensitive skin, which is easily damaged [51,52].

Despite the large variance in stocking density in the trailer, no effect on the incidence of skin damage was reported, which supports the findings of Guise et al. [40]. In our study, the maximum stocking density (0.425 m^2^ per 100 kg) was respected in all transports [20]. However, this standard is not always applied in commercial conditions, as reported by Christensen et al. [53] and more recently by Van de Perre [54]. The high variation in transport density results in studies which conclude an effect of stocking rates during transport [30]. On the other hand, more space (densities higher than 0.35 m^2^ per pig) can lead to more skin damage due to an unadjusted driving style [55]. In contrast with our study, Guàrdia et al. [8] detected an effect of season and stocking density on the incidence of skin damage. The maximum stocking density in their study was much higher (416 kg/m^2^) as well as the variance between minimum (156 kg/m^2^) and maximum, which might be the explanation for the effect they found. The seasonal effects in the study of Guàrdia et al. [8] might be explained by more extreme weather variables than in Belgium. Although the effect of transport time is shown [32,33], we did not find any relationship between the transport time and skin damage, probably due the smaller variance in transport time in our study. 

### 4.3. Source of Skin Damage

The assessment of skin damage at the slaughter line not only helps to determine the number of marks on the carcasses, but also may allow one to recognise the source (fighting, rough handling and mounting) according to the anatomical location [13]. Damage on the head and shoulder area are caused by fights connected with social ranking [14]. Damage on the back are caused by mounting, by scratches with the claws [26] and rough handling [13]. Mounting behaviour usually occurs at overcrowded conditions in the pen. These facts suggest that, in our study, 30.98% of the skin damage (shoulder part; score > 1) would be caused by fighting during transport. Furthermore, mounting/handling during the transport procedure is responsible for 28.26% skin damage on the middle and 11.78% on the ham.

### 4.4. Skin Damage Method

The most severe skin damage score is only detected in a few cases. Therefore, it can be discussed if a 4-point scale is useful and if a 3-point scale is enough to evaluate skin damage in practice. However, we would argue that the 4-point scale should be kept. While there are few occurrences, it is necessary to identify and distinguish the extreme cases, determined by score 4, especially when developing critical control points. According to the relations with conditions during and after transport in our and previous studies [56], and the relation with physiological measures [34,56], we confirm that the skin damage score is a reliable indicator to monitor the welfare of pigs on the day of slaughter. The skin damage score can be used as a tool to determine critical control points in the transport process, i.e., the loading at the farm till slaughter, and to improve animal welfare at the transport and the slaughterhouses by using thresholds that can be adjusted or tightened in the time. These recordings are easy to implement but are labour intensive and therefore a barrier to the continuous documentation of animal welfare. The development of tools for the continuous automatic monitoring of skin damage is therefore required [57].

## 5. Conclusions

Sex, regrouping pigs during loading, the ambient temperature and humidity during transport, the combination of transport factors, and lairage time all affect skin damage, especially to the shoulder region. The higher incidence of skin damage in the shoulder region, compared to the middle or ham suggests that the damage was caused by fighting in the preslaughter period. Mixing unfamiliar pigs for transport or in lairage is a critical control point and therefore assessing skin damage on the shoulder in the slaughter line is recommend. The online monitoring of skin damage on carcasses should be included in routine inspection procedures, as a complementary tool to identify critical points along the transport and slaughter procedure and to optimize welfare-friendly handling and, ultimately, the financial rewards. In practice, the common preslaughter of mixing of pigs and rough driving during transport can be re-educated via training programmes, which are now mandatory for animal transporters and handlers in the EU.

## Figures and Tables

**Figure 1 animals-10-00575-f001:**
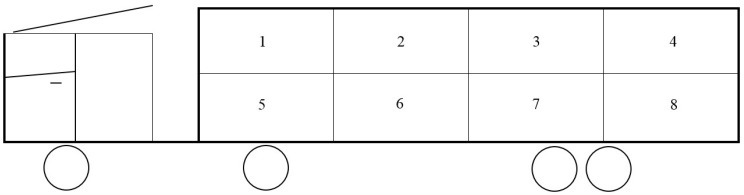
Schematic presentation of the trailer with the numbered trailer pens. Pigs transported: *n* = 786 in trailer pen 1; *n* = 1122 in trailer pen 2; *n* = 1127 in trailer pen 3; *n* = 896 in trailer pen 4; *n* = 23 in trailer pen 5; *n* = 183 in trailer pen 6; *n* = 334 in trailer pen 7; *n* = 36 in trailer pen 8.

**Figure 2 animals-10-00575-f002:**
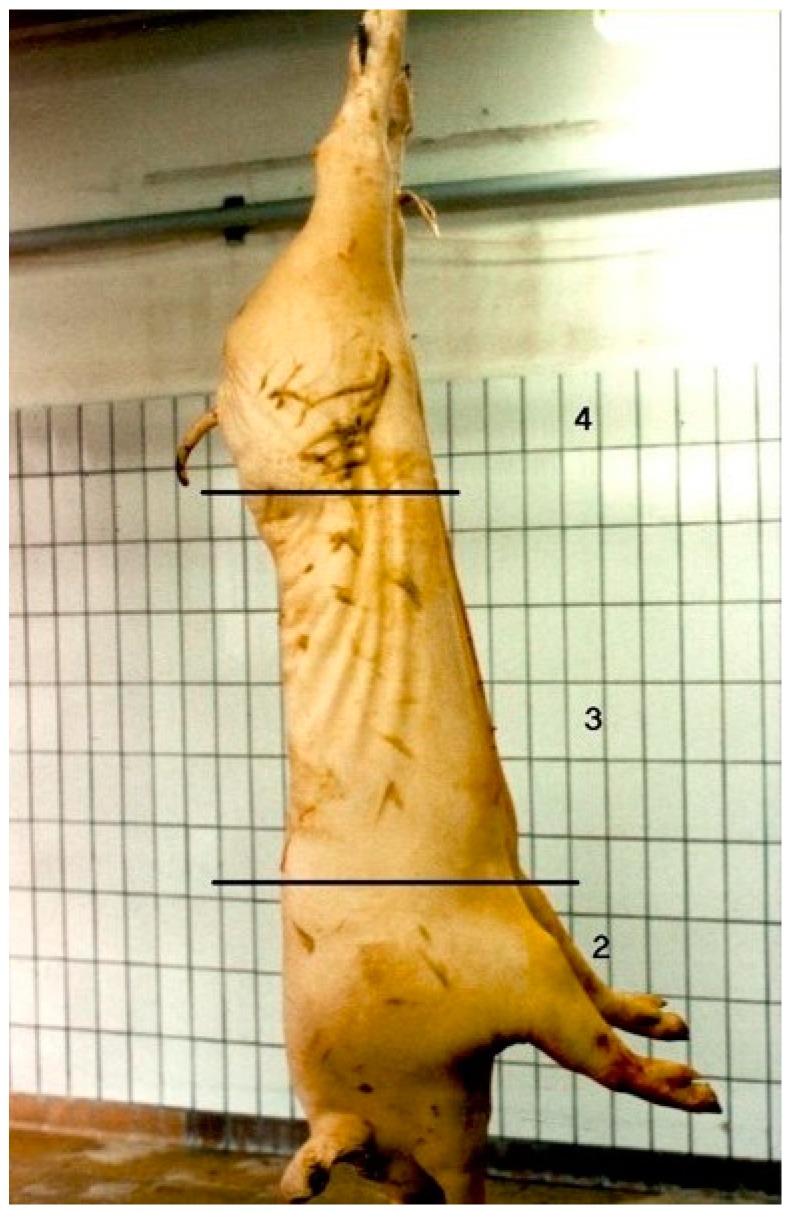
Scoring skin damage per part of the carcass, i.e., shoulder, middle and ham, starting at score 1 = no damage and 4 = extreme damage [14]. The lesions shows brown because this picture was made after singeing in contrast with the used skin damage scoring technique which was before singeing.

**Table 1 animals-10-00575-t001:** Used transport combinations during the monitoring.

Combination	Truck Number	Trailer Number	Driver Number	Floor Space per Pen (m^2^)	Number of Transported Pigs
1	1	1	1	6.63	2903
2	1	1	2	6.63	386
3	2	1	1	6.63	346
4	2	1	3	6.63	419
5	3	2	4	4.98	453

**Table 2 animals-10-00575-t002:** Weather, transport, and carcass variables.

Variable	Mean	SD	Median	Maximum	Minimum
Weather ^a^					
Dry air temperature, °C	12.9	6.6	12.9	34.0	0.2
Relative humidity, %	81.8	10.2	85.0	95.0	15.0
THI	54.8	10.3	53.9	86.7	35.3
Wind velocity, km/h	14.9	9.7	13.0	60.0	0.0
Transport (*n* = 128)					
Transport time, min	98.7	15.9	95.0	160.0	80.0
Unloading time, min	28.0	18.5	23.0	95.0	4.0
Lairage time, min	83.8	34.0	85.0	165.0	2.0
Loading density in trailer, kg/m^2^	192.0	15.0	190.0	235.0	125.0
Carcass variables (*n* = 4507 pigs)					
Live weight, kg	106.0	5.3	105.0	136.0	88.0
Carcass weight, kg	82.5	6.9	82.3	106.0	62.1
Ham width, mm	196.0	10.7	196.0	231.0	156.0
Ham angle, °C	48.9	9.1	49.0	73.0	18.0
Loin width, mm	127.0	7.6	127.0	157.0	102.0
Lean meat, %	59.4	2.6	59.6	66.7	45.2

^a^ Weather characteristics registered during each transport.

**Table 3 animals-10-00575-t003:** Incidence of the skin damage of the carcasses of 4507 pigs was scored.

	Incidence (%) on Carcass Part			
Score ^a^	Shoulder	Middle	Ham	Highest carcass score ^b^
1	69.02	71.74	88.22	53.78
2	20.32	21.89	9.22	31.36
3	8.34	5.60	2.21	12.15
4	2.32	0.77	0.35	2.71

^a^ In the slaughter line, skin damage was visually assessed in different parts of the carcass, i.e., shoulder, middle, and ham, using a 4-point scale with 1 = no damage and 4 = extreme damage. ^b^ The highest scoring body part determined the ultimate score.

**Table 4 animals-10-00575-t004:** Factors influencing the incidence of skin damage. The dependent variables are shoulder, middle, and ham damage. For continuous variables, the direction of the relationship (+ or −) is shown.

Variable	Shoulder Damage	Middle Damage	Ham Damage
Sex	*	*	
Transport season			
THI			
Wind velocity	** −	*** −	** −
Regrouping ^a^	***	*	
Transport combination ^b^	*	*	
Transport compartment ^c^			**
Transport density			
Transport time			
Unloading time			
Lairage time		* +	* +
Ham angle	** +		

* *p* < 0.05, ** *p* < 0.001, ****p* < 0.0001. ^a^ Two treatments were tested: unmixed condition and mixed before transport. ^b^ Used transport combinations shown in Table 1. ^c^ The pigs were transported in a two-tier trailer with 4 compartments per tier.

**Table 5 animals-10-00575-t005:** Frequencies (*n* = 4507) of the effect of sex, regrouping, transport combination, and transport compartment on skin damage. The dependent variables are shoulder, middle, and ham damage. Shown *p*-values are per carcass part (shoulder, middle and ham).

Variable	*p* Shoulder	*p* Middle	*p* Ham	Level	Skin Lesion Score ^a^	Shoulder Damage (%)	Middle Damage (%)	Ham Damage (%)
Sex	*p* = 0.0035	*p* = 0.0038	N.S.	Castrated male	1	71.41	74.25	89.10
					2	19.19	19.73	8.73
					3	7.36	2.27	1.77
					4	2.04	0.75	0.40
				Female	1	66.67	69.20	87.39
					2	21.48	24.06	9.63
					3	9.28	5.95	2.66
					4	2.57	0.80	0.31
Regrouping	*p* < 0.0001	*p* = 0.0267	N.S.	Unmixed condition	1	86.69	80.77	93.20
					2	9.76	16.86	6.80
					3	3.25	2.07	0.00
					4	0.30	0.30	0.00
				Mixed before transport	1	67.76	74.14	87.76
					2	21.01	22.16	9.38
					3	8.73	5.88	2.37
					4	2.49	0.82	0.38
Transport combination ^b^	*p* = 0.0064	*p* = 0.0475	*p* = 0.0441	Combination 1	1	70.31	71.10	87.98
					2	18.05	21.60	9.03
					3	8.75	6.27	2.51
					4	2.89	1.03	0.48
				Combination 2	1	81.69	88.52	96.17
					2	11.20	9.29	3.01
					3	5.19	1.91	0.82
					4	1.91	0.27	0.00
				Combination 3	1	67.78	72.70	88.34
					2	27.30	25.15	10.74
					3	5.21	2.15	0.92
					4	0.00	0.00	0.00
				Combination 4	1	57.04	69.35	81.66
					2	30.40	22.86	14.32
					3	10.30	7.04	3.52
					4	2.26	0.75	0.50
				Combination 5	1	61.34	67.59	90.74
					2	28.34	25.93	8.33
					3	9.49	6.48	0.93
					4	0.93	0.00	0.00
Transport compartment	N.S.	N.S.	*p* = 0.0004	Compartment 1	1	71.76	73.54	88.63
					2	18.19	20.23	9.15
					3	7.51	4.96	1.87
					4	2.54	1.27	0.36
				Compartment 2	1	68.29	70.60	88.63
					2	21.58	23.71	9.15
					3	8.08	5.06	1.87
					4	2.04	0.62	0.36
				Compartment 3	1	68.02	69.52	88.02
					2	19.56	23.44	10.04
					3	9.78	5.99	1.67
					4	2.64	1.06	0.26
				Compartment 4	1	74.44	75.67	90.63
					2	16.07	19.64	6.92
					3	7.48	4.24	2.12
					4	2.01	0.45	0.33
				Compartment 5	1	69.56	73.91	86.95
					2	21.74	21.74	8.70
					3	8.70	4.35	4.35
					4	0.00	0.00	0.00
				Compartment 6	1	60.11	88.52	84.70
					2	32.79	9.29	12.57
					3	6.01	1.91	2.73
					4	1.09	0.27	0.00
				Compartment 7	1	54.79	58.98	75.75
					2	30.84	28.14	16.17
					3	10.78	12.28	7.19
					4	3.59	0.60	0.90
				Compartment 8	1	74.99	74.99	86.11
					2	13.90	16.67	8.33
					3	8.33	5.56	5.56
					4	2.78	2.78	0.00

^a^ Skin lesion score: 4-point scale with 1 = no damage and 4 = extreme damage. ^b^ Used transport combinations shown in Table 1. N.S.: not significant.

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
