# Peer review of "Effects of Transport and Lairage on the Skin Damage of Pig Carcasses"

_animals, 2020, doi:10.3390/ani10040575_

Round 1

Reviewer 1 Report

Transport is a large and integral part of today's livestock industry. However, it is also considered the most stressful event for animals including pigs prior to slaughter. Animal welfare concerns include the potential for the animals to experience stress, injury, fatigue, mortality and morbidity that may be caused by numerous factors including poor handling and mixing with unfamiliar animals. Therefore, evaluation of the incidence of skin lesions in pigs and analysis of the effects of different factors under commercial conditions is of high importance.

I consider the paper to be a valuable contribution to the current knowledge in this area.

Specific comments:

  1. „Pigs were fasted 18 h before transport.“ Why? The EU legislation do not allow animals to be fasted for more than 12 hours before slaughter?
  2. „In conclusion, the incidence of skin lesions is influenced by many factors and is an indicator of the level of welfare exercised during transport and slaughterhouse conditions.“ The conclusion is way too general. Basically, this is what most papers related to animal transport say. I would expect a novel conclusion derived from your results.

Reviewer 2 Report

Overall this was a well-designed study with an impressive population size. The paper was clear and enjoyable to read. Please consider the following points to further strengthen the paper.

Line 13: at the slaughterhouse

Line 51: Is the farmer ultimately getting hit with financial losses due to poor transport and handling at slaughter? How are transport handlers being held accountable for losses they may be responsible for?

Line 89: Table 1, extra spacing between ‘number’ and ‘of’ in the last column header

Line 104: Could you further explain the rationale for including the combination effect in the model as opposed to including each factor independently?

Line 103: Why so few groups of unmixed pigs? I’m guessing it was due to logistical reasons and mixed groups are probably more representative of industry practices, however it might have helped to further tease apart factors not related to aggression that lead to lesions.

Line 117: Bummer this couldn’t have been more balanced. Would it be typical for the lower level to remain mostly empty during transport?

Line 134: Were the pigs evaluated for lesions separately by both scorers? Or did the evaluation team come to a conclusion on the score together? Please elaborate.

Line 137: It would be great to have a bit more detail on how the scores were determined. For instance, were you most interested in number of lesions, severity of lesions, or proportion of body covered in lesions? What would distinguish between a score of a 2 versus a 3?

Line 138: It doesn’t appear that lesions present prior to loading/transport were accounted for. While I would expect little aggression occurring in their home pen, there is always the possibility that pigs were fighting in these groups potentially leading to skin lesions at slaughter that would not be due to transport or lairage conditions. Thoughts?

Line 166: “Large variations (minimum versus maximum, Table 2) in transport, unloading and lairage time is determined.” Consider rewording for clarity. ‘were determined’ instead?

Line 183: Table 4, potentially consider the addition of lines in the table to ease readability.

Line 230: Could you further elaborate on why more thermally comfortable pigs would accumulate fewer lesions? Would they be more likely to lie down during transport and thus avoid injuries?

Line 287-288: Citation?

310-312: Can you really conclude this?

Line 316: I would argue that the 4 point scale should be kept. While there are few occurrences, it is necessary to identify and distinguish these extreme cases, especially when developing critical control points.

Reviewer 3 Report

Dear Authors,

Thank you for your very nice draft. I recognized the work you already put in this draft. It was easy to read and the main aspects of your work were clear. Some aspects wasn't clear for me, that's why I made suggestions for improvement. I really hope my comments and suggestions will help.

Best wishes.

In General:

I suggest using "skin damage" for your study, instead of “skin lesions”. The words "skin lesions" are more famous for husbandry and the assessment in live pigs. In addition, scoring methods for live pigs differs from the ones you used. Please make it the same for your tables and figures.

Simple Summary:

Line 12: Why are effects not clear? Please specify.

Introduction:

Please mention the impact of lairage on skin damage in your introduction. Now, you only refer to the impact of transport on animal welfare and economic losses. But in your title you also mention “lairage”. You also should cite more studies, that already investigated some effects/factors on skin damage during transport and lairage, but even not all, to make more clear why your study has a new approach. Especially for line 59 – 65, I missed the references.

Line 45-46 are a repetition of lines 54 - 58. Please combine in terms of content.

Line 51- 53: I missed the reference in your sentence.

Line 48- 49: [Compliance with these...] – it was not clear for me, why do you mention development of on-site tools for continuous monitoring of welfare of slaughter pigs. Can you specify this? How does it refer to your study?

Please write some sentences, why you were scoring three body parts (shoulder, middle, ham) instead of the whole carcass. Did you expect differences in incidences on these parts? What was the aim to assess skin damage on shoulder, middle and ham, separately?

Material and Methods:

Please arrange the text in more subchapters and think about a new structure. Now, the description of your data collection is not easy to get from the text and this makes it a bit hard to follow your results, e.g.in table 2.

I suggest following structure:

2.2. Animal and housing

2.3. Transport and Slaughterhouse (or Lairage)

Here, only describe transport combinations as you already did in 2.2. and also housing facilities in lairage.

2.4. Data collection

2.4.1 Weather parameters during transport

In which season do you transported the pigs?

2.4.2 Time sampling during transport and lairage

2.4.3. Carcass variables

2.4.4. Skin damage

2.5. Statistical analysis

Line 80 – 82: In 2.4. You wrote that you are able to distinguish older from fresh lesions by colour. Which affects has the handling in husbandry on your investigation? Did you mean the handling in lairage?

Line 121 – turn over to next page.

Line 137: Please describe all 4 scores, 1 = no damage, 2 = …, 3 = … and 4 extreme damage. The reference you quote is very hard to get.

Line 139: Please quote the origin reference Gracey 1986.

Line 140 – 143: Does not fit to measurements of skin damage. It is more a carcass variable.

Line 141: Schlachtkörper

Line 146: Figure 2: It is not clear for me, what you mean with the numbers 2/3/4 within the picture. If you mean the scores, it is confusing because the skin damage shown, have more brown lesions than red or yellow ones. Maybe you find a picture that illustrates the descriptions within your text in 2.4 better.

Statistical analyses

Did you explain the recording of the gender?

Line 156: Please be consistent in using words. Use weather parameters or characteristics (see table 2.), but not both terms for the same variables.

Results

Table 2: Please check the number of your decimal digits. There are differences in the number of decimal digits in some lines.

Table 3: Can you explain in Material and Methods, how do you calculated the highest carcass score and why? And did you calculated an ultimate score?

Discussion

Line 230 – 232: I could not follow your argument. How refers thermoregulation to the incidence/prevalence of skin damage on carcasses. Can you explain this more detailed?

Line 252: Can you put some information on your drivers’s qualities in your transport combinations. This is missing in your study.

Line 286: I did not really get, what do you mean with conformation and how does is refers to skin damage on the ham. Can you specify this?

Line 320: Which critical control points do you mean? Can you explain it more detailed?

Line 323: Use “skin damage” instead of “animal welfare”. You only refer to skin damage as animal-based indicator in your study. Thus, skin damage on carcasses is one possible indicator to assess animal welfare.

Conclusions

Line 328. Why do you consider scoring of skin damage on shoulder as measurement of animal welfare? Skin damage on shoulder is a result of fighting for social hierarchy or antagonistic behaviour, but not a fight for resources. Did you mean, that mixing unfamiliar pigs for transport or in lairage is a critical control point and that’s why you recommend to asses skin damage on shoulder in the slaughter line?

Round 2

Reviewer 3 Report

Dear Authors, thank you very much for your new Vision. I really like it. I still found some very Little mistakes, but no big mistakes.

Thank you and best regards.

19: Damage instead lesion

29: Damage instead lesion

31: Damage instead lesion

210: Damage instead lesion

210 page 10 of 16: 4.1.: Damage instead lesion

321 4.4. Damage instead lesion

322: Damage instead lesion

328: Damage instead lesion

329: Damage instead lesion